# The Needs for Developing Experiments on Reservoirs in Hantavirus Research: Accomplishments, Challenges and Promises for the Future

**DOI:** 10.3390/v11070664

**Published:** 2019-07-19

**Authors:** Sarah Madrières, Guillaume Castel, Séverine Murri, Johann Vulin, Philippe Marianneau, Nathalie Charbonnel

**Affiliations:** 1Agence Nationale de Sécurité Sanitaire de L’alimentation, de L’environnement et du Travail, Laboratoire de Lyon, Unite Virologie, 69007 Lyon, France; 2Centre de Biologie Pour la Gestion des Populations (CBGP), INRA, CIRAD, IRD, Montpellier SupAgro, Univ. Montpellier, 34000 Montpellier, France

**Keywords:** hantavirus, experimentations, reservoirs, transmission, spill-over, persistence, immunity, super-spreader, evolutionary ecology, virology

## Abstract

Due to their large geographic distribution and potential high mortality rates in human infections, hantaviruses constitute a worldwide threat to public health. As such, they have been the subject of a large array of clinical, virological and eco-evolutionary studies. Many experiments have been conducted in vitro or on animal models to identify the mechanisms leading to pathogenesis in humans and to develop treatments of hantavirus diseases. Experimental research has also been dedicated to the understanding of the relationship between hantaviruses and their reservoirs. However, these studies remain too scarce considering the diversity of hantavirus/reservoir pairs identified, and the wide range of issues that need to be addressed. In this review, we present a synthesis of the experimental studies that have been conducted on hantaviruses and their reservoirs. We aim at summarizing the knowledge gathered from this research, and to emphasize the gaps that need to be filled. Despite the many difficulties encountered to carry hantavirus experiments, we advocate for the need of such studies in the future, at the interface of evolutionary ecology and virology. They are critical to address emerging areas of research, including hantavirus evolution and the epidemiological consequences of individual variation in infection outcomes.

## 1. Introduction

Zoonoses, i.e., infectious diseases that are spread between animals and humans, are an increasing and major threat to public health. Among them, about 75% are transmitted from wildlife to humans [1], which shows the need to develop research on wild, non-model animals, to identify reservoirs of zoonotic pathogens, assess the different modes of pathogen transmission and characterize reservoir/pathogen interactions. In such research, the combination of correlative approaches based on field surveys with experimental studies is critical to make and test hypotheses with regard to these ecological, epidemiological and evolutionary issues.

Since their discovery in the late 1970s, hantaviruses (family Bunyaviridae, genus Orthohantavirus) have been detected worldwide, in a broad spectrum of wild animal reservoirs, including rodents, bats and soricomorphs (see for a review [2]). About one half of the 36 species of hantaviruses recorded are zoonotic agents [3], responsible for renal or respiratory diseases in humans (hemorrhagic fever with renal syndrome or hantavirus cardiopulmonary syndrome). The case-fatality rate in humans is highly variable depending on the hantavirus considered, but it can reach 15% (e.g., Hantaan virus (HTNV), [4]). By contrast, hantaviruses are usually considered to be asymptomatic in their specific reservoirs [5].

Hantaviruses have therefore been the object of a large array of clinical, virological, ecological and evolutionary studies. In particular, many experimental studies have been conducted to identify the mechanisms leading to pathogenesis in humans and to develop treatments of hantavirus diseases. In vitro approaches, based on reservoir-derived cell lines, have been carried out to study hantavirus-reservoir interactions (e.g., hantavirus entry and replication, reservoir host immune response to hantavirus infections and hantaviruses immunity escape mechanisms, see for a review [6]). Despite recent improvements, in vitro models can still not reproduce the systemic interactions that occur between organs, within individuals, during hantavirus infections [7]. It is therefore critical to carry experiments on animals to study hantavirus–reservoir interactions.

In parallel, hantavirus research has been developed on laboratory animal models for the study of hantavirus-reservoir interactions, for the identification of the mechanisms leading to pathogenesis in humans and for the design of vaccines and antiviral therapeutics [8]. The main achievements gathered from mice, Syrian hamsters and non-human primates have been reviewed in Smith et al. [9]. They mostly concerned research areas dedicated to the understanding of human pathogenesis and the elaboration of vaccines and antivirals.

Therefore, hantavirus experimental studies carried out in the natural reservoirs are critical to provide knowledge and test hypotheses on hantavirus-reservoir interactions. These studies are still scarce considering the diversity of hantavirus/reservoir pairs described in the wild, and the wide range of issues that need to be addressed from the eco-evolutionary and virological perspectives. This review therefore provides a comparative assessment of hantavirus-reservoir interactions among a wide species range. As such, it enables to emphasize the gaps that need to be filled in the future, both from the biological hantavirus/reservoir pairs considered and the scientific topics addressed.

## 2. Insights into Infection Dynamics of Hantaviruses and Their Impact on Reservoir Fitness

For many hantaviruses, including Puumala (PUUV), Sin nombre (SNV), Seoul (SEOV), Hantaan (HTNV) and Black Canal Creek (BCCV) viruses, experimental infections have been performed in laboratory conditions, outdoor enclosures or using capture-mark-recapture (CMR) studies on reservoir hosts. These experiments have been critical to describe the infection dynamics of hantaviruses in their reservoirs, to analyze epidemiological features of these infections (transmission, pathogenesis), and to learn more about the mechanisms mediating viral persistence in reservoirs and maintenance in natura [10].

### 2.1. Dynamics of Hantavirus Infection in Reservoirs

#### 2.1.1. Viremia, Sites of Hantavirus Replication and Persistence

Experiments have been indispensable to reveal that the dynamics of hantavirus infection in their reservoirs happens in two stages: an acute phase, followed by a chronic/persistent phase, (e.g., [8,11,12,13,14,15]).

During the acute phase, experiments have shown that a transient viremia usually occurs between seven and 15 days (e.g., in bank voles and rats, [11,12]), but in some cases viral RNA can be detected earlier in sera, such as at three days post-infection (dpi) during PUUV infection of bank voles [16]. For some hantaviruses, viral RNA is detected at later points. At 21 dpi, circulating virus was detected in 20% of cotton rats infected experimentally with BCCV [14], and SNV was detected in the blood of all deer mice infected, at 21 and 28 dpi. Exceptionally, Kariwa et al. [17] detected viral genome from one to 184 dpi in the serum of Wistar rats infected at one day-old by SEOV.

The different studies that have been carried out have highlighted that the transition between these phases could occur at different times according to the hantavirus/reservoir pairs studied. Generally, this transition occurs during the first two–three weeks of infection [18], but in some cases, it occurred later. For BCCV in hispid cotton rat (*Sigmodon hispidus*), it has been shown that the chronic phase occurs around 50 dpi [14] while it occurs between 60 and 90 dpi for SNV in deer mice *Peromyscus maniculatus* [15].

Hantavirus experimental infections, by controlling time since infection and by allowing the simultaneous analyses of several organs, have enabled to discriminate the sites for viral replication and the sites for viral maintenance during persistent infection. The viral replication seems to occur systematically all along the acute phase of infection. Indeed, hantaviruses are found in many organs and tissue, including kidneys, salivary glands, liver, heart, brown fat, testicle, adrenal, spleen (e.g., BCCV, [14]), skin [19] and more frequently in lungs with higher viral loads (e.g., [12,20,21], so that this latter organ is considered to be the primary site of replication for hantaviruses. The type of infected cells in each organ has not always been identified, but it had been shown that in lungs, alveoloar macrophages and epithelial cells can be infected [18,22] and that Kuppffer and endothelial cells are infected in the liver [12,15].

Long-term experimental studies have revealed that viral RNA could remain detectable during a long period after hantavirus infection. Infectious virus was detected in lungs and liver until 270 dpi during infection of bank voles (e.g., [12]). Botten et al. [15] found that some laboratory- inoculated deer mice had detectable SNV RNA in lungs and heart for as much as 217 dpi. These experimental studies have highlighted that these patterns of viral distribution during the persistent phase were highly variable between infected individuals or throughout the course of the persistent phase. It is therefore likely that periodic episodes of viral recrudescence can take place during persistent infection. The reasons for these bursts have not been investigated, in laboratory conditions or in the wild.

#### 2.1.2. Modes and Kinetics of Hantavirus Shedding

Hantaviruses have been found in a various array of excreta/secreta of their infected reservoirs (urina, saliva and feces, e.g., in PUUV infected bank voles, [23]). However, experimental studies have reported a large variability of excretion patterns depending of the hantavirus/reservoir pair studied and the infection protocols used. The earliest transmission of PUUV to other bank voles (5 dpi) was reported by Gavrilovskaya et al. [13]. Similarly, the occurrence of hantavirus shedding in early time of infection (7 dpi) has been described in a number of hantavirus/reservoir pair experimental studies conducted in laboratory conditions ([11] for HTNV/*Apodemus* sp.; [24] for SEOV/rats; [14] for BCCV/sigmodon rats) or in CMR surveys ([25] for PUUV; [26] for SNV). However, a number of other studies, sometimes performed on the same hantavirus/reservoir pairs, have found that shedding occurred a little bit later (15 dpi, for example for PUUV/bank voles [12] or for SNV/deer mice [27]).

Moreover, most studies have reported that only periodic shedding seemed to occur during the persistent phase of hantavirus infection ([12,15,22,23,28,29]). However, Voutilainen et al. [30] have recently shown from a two-year CMR survey that bank voles shed PUUV even eight month after seroconversion, with only a slight decline.

The reasons for this variability in shedding patterns have not been explored. The mode of transmission, the quantity of virus inoculated or some individual factors could mediate these differences (Table 1). As a matter of fact, the influence of experimental infection design on the patterns of shedding is evidenced by studies in which shedding could not even be reproduced. Hence, no viral RNA was detected in excreta/secreta of bank voles infected with PUUV [16], while infectious PUUV was detected in oropharyngeal secretions and feces of PUUV infected bank voles colony [12]. More broadly, it is important to keep in mind that detection of viral RNA between studies might rely on molecular tests with potential significant difference in sensibility, which would prevent the comparison of the results obtained. 

### 2.2. Hantavirus Infection’s Impact on Reservoirs

For a long time, hantavirus infections have been considered as being asymptomatic in natural reservoirs. Most experiments carried out on natural reservoir species in laboratory conditions have revealed the absence of disease (e.g., [14,19,22]) or clinical signs (behavior or physical modifications, e.g., [14]) during hantavirus infections of a large array of hantavirus/reservoir pairs. Moreover, many histopathological analyses did not detect any change in host tissue during hantavirus infection [15,19]. The same results were gathered from outdoor enclosure experiments and ecological surveys (e.g., [8,25]).

Similarly, experimental hantavirus infections did not seem to affect reservoir life history traits, including survival (e.g., [43,47,70]), reproduction (e.g., [51]) or growth rates [70].

However, some more recent, long-term studies have shown that hantavirus infections did have an impact on reservoirs fitness, at least for some individuals. Several surveys reported the negative impact of hantavirus infections on reservoirs survival (e.g., [28,56,57,59,60], 2001) or on the reproductive output of female reservoirs (e.g., [61]). Moreover, Fulhorst et al. [29] found that about one third of Cano Delgadito Virus (CDGV) infected *Sigmodon alstoni* appeared lethargic (inactive, ruffled haircoat) between 10 and 15 dpi. Easterbrook et al. [41] observed hemorrhage and edema in the lungs of SEOV infected rats, and some neurological damages were also observed after experimental inoculation of SEOV to two days-old rats [19].

With regard to these very conflicting observations, we advocate for a deeper analysis of the impact of hantavirus infection on natural reservoirs, and on the factors that can make infection vary from asymptomatic to impairing ones, be they due to genetics, environmental features or to their interactions.

## 3. Hantavirus Transmission: Modes, Dynamics and Spill-Over

### 3.1. Modes of Hantavirus Transmission

Understanding how hantaviruses are transmitted within populations of their reservoirs is critical to set up effective prevention strategies [73]. The majority of knowledge pertaining to hantavirus transmission comes from well-studied reservoir systems and mainly PUUV in bank voles and SNV in deer mice [74].

#### 3.1.1. Routes of Inoculation

For experimental infections, different routes of inoculation have been tested successfully. In most cases, the inoculation route is chosen to mimic as much as possible the natural mode of transmission. Lee et al. [11] have shown that intramuscular (IM), subcutaneous, intraperitoneal (IPe), intranasal and intrapulmonary (IPu) inoculations of HTNV virus in *Apodemus* sp. resulted in successful infection. However, the intrapulmonary route gave the highest infectivity index and intranasal and oral routes were least efficient [11]. Very recently, Warner et al. [48] have shown that IPe infection appeared to be a more reliable infection route compared with IM infection for SNV in *Peromyscus maniculatus.*

Different sources of contamination have also been tested. Relative importance of saliva, urine and feces in transmission of PUUV between bank voles and why levels of virus change over time in different excretions are not known. All three excretions can transmit virus to other bank voles when administered intranasally, which suggests that all three excretion pathways can function as natural transmission routes between bank voles [23].

#### 3.1.2. Vertical versus Horizontal Transmission

It is generally admitted that hantaviruses are not transmitted vertically from dam to offspring. This was demonstrated in a number of studies, including a model of laboratory mice infected by HTNV [75] and experimental infections of deer mice with SNV (e.g., [27]). This hypothesis is also strongly supported in SNV naturally infected *Peromyscus maniculatus* [62]. In contrast, the data of Hutchinson et al. at least suggested the possibility of in utero transmission of BCCV [51]. Therefore, although these events of vertical transmission have rarely been detected, they deserve further attention to better evaluate their frequency and their efficiency, at least in the BCCV/*Sigmodon hispidus* pair.

#### 3.1.3. Direct versus Indirect Transmission

The exact mechanism(s) by which each hantavirus is transmitted within its reservoir host community is still incompletely understood. The role of indirect (aerosols) and direct (physical contacts, bites, grooming, shared nesting) transmission modes in hantavirus dynamics, as well as the main route of hantavirus shedding have been discussed extensively and the results have been rather conflicting depending on the system studied (Table 1, e.g., [5,13,14,20,21,24,27,38,69,76]).

The role of intraspecific aggression has been essentially studied by observing wounds. Evaluation of the potential for hantaviral transmission by wounding associated with aggressive interactions was first described by Glass et al. [71], who found significant association between animal’s serological status and the presence of wounds.

SNV and Andes virus (ANDV) are commonly shed in saliva and have rarely been identified in urine and feces [26,27,69], suggesting that intense contact is primarily responsible for intraspecies transmission [74]. In enclosures, SNV-infected deer mice have a higher wound frequency than uninfected deer mice, clearly demonstrating an association between wounding and infection [66]. This was also shown in another study in wild population [77]. Very recently, Warner et al. [48] confirmed that direct contact between deer mice was the major driver of SNV transmission. However, it was also shown that non-aggressive contact patterns (sniffing, touching, mutual exploration) could also result in direct transmission of ANDV to *Oligoryzomys longicaudatus* through saliva [69].

Furthermore, indirect inhalation of aerosolized virus in the environment has been considered as the dominant mode of hantavirus transmission due to the ease with which the virus can pass between reservoirs [74]. Events of hantavirus transmission resulting from indirect contacts between reservoirs have been demonstrated (e.g., PUUV [13], SEOV [78], [24] and HTNV [11]). Moreover, Kallio et al. [32] have underlined that this indirect transmission occurred over an extended time period. PUUV remained infectious in bank vole cage beddings for 12–15 days at room temperature after removal of the infected animals. Obviously, for indirect transmission the stability of the virus in environment is crucial. However, the influence of environmental conditions (temperature, humidity, UV radiation) remains to be assessed more deeply in natural settings. Modeling studies have already predicted that voles can be infected several days after disappearance of the infectious excreting voles in a wet soil environment [79]. This is in agreement with another study that suggested that low winter temperatures and high soil moisture favor the persistence of PUUV in the environment [80].

#### 3.1.4. Transmission by Ingestion

Intranasal inhalation of virus may also involve ingestion, which may also be a viable route of infection. Furthermore, ingestion could occur when several bank voles share a common food source and infection via the alimentary tract might be a possible route of contamination. Up to now, some evidence has only been provided by experiments conducted on laboratory animal models. Hooper et al. [52] have shown that ANDV is infectious to hamsters when administered by intragastric injection and suggested that ingestion of contaminated material might be a mode of transmission to humans. This way of contamination has also been demonstrated with PUUV in Syrian hamsters [37].

Altogether, experimental studies have identified a large array of feasible modes of transmission, in laboratory or environmental conditions. Nevertheless, the relative importance of these pathways in natura and the potential temporal variations of the predominant routes of infections throughout the course of the infection remain unclear and could depend on hantavirus/reservoir pairs.

### 3.2. Environmental Factors Influencing Hantavirus Transmission

Several environmental features that could influence hantavirus transmission have been tested using experiments carried out in outdoor enclosures or through ecological surveys. One of the main issues addressed concerned the potential impact of small-mammal diversity on the encounter rates between infected and susceptible individuals (the “encounter reduction” mode of the “dilution effect”). This dilution effect has been successfully tested by manipulating the species richness and population densities of reservoir and non-reservoir species in experimental plots at edges of small forest fragments in Panama [72]. Authors have evidenced a clear negative effect of high species diversity on both the abundance of reservoir hosts and their infection prevalence with hantavirus. In a more recent study in Northwest Mexico, the same group has shown that not only reservoir species richness or diversity were important but also the identity of species and the composition of the assemblages in the outdoor enclosures [68]. Correlations between high reservoir diversity and low rates of pathogen transmission or disease risk have been found for several other hantavirus/reservoir pairs, including PUUV/bank voles [57] or deer mice/SNV [64].

Seasonality is another major factor influencing hantavirus transmission that has been investigated using outdoor enclosures and CMR ecological surveys. These experiments have highlighted that breeding season was favorable to direct transmission especially in young males, as these experienced more aggressive behavior (e.g., in SEOV infected rats, [20,63]). By contrast, Voutilainen et al. [30] have shown that PUUV transmission rate increased in autumn, outside the breeding season, when bank vole abundance increased in natural populations. Their results corroborated the idea that aggression is not an important factor affecting PUUV transmission in boreal bank voles, and that the key routes of transmission are likely to vary between hantavirus/reservoir pairs [30]. Altogether, these findings highlighted the strong interactions that might exist between the predominant modes of hantavirus transmission, their temporal variations and the influence of environmental features. For example, shedding in saliva might be more efficient for virus transmission in males living in a high-density area during mating season, in reservoir species exhibiting aggressive behaviors. By contrast, shedding in feces may play a more dominant role in transmission in a low-density area during fall or winter as the virus will remain in more favorable environmental conditions [23].

### 3.3. Transmission to Non Reservoir Hosts

Although each hantavirus seems to be associated with one reservoir host species, several surveys conducted in natural populations reported spillover events, i.e., hantavirus infected animals that are not natural reservoirs of a given hantavirus [81]. This process could potentially contribute to the evolution and maintenance of the pathogen in natura. Thus, it is very important to know whether hantaviruses, which are associated with specific rodent host species, could infect, or cause persistent infection in closely-related animal species that are hosts—or not—to a related hantavirus. To investigate reservoir host specificity and species restriction of new world or old world hantaviruses, several experimentations have been carried out on colonized and wild trapped animals.

In a large experiments that included several old world hantaviruses (PUUV, Tula Virus (TULV), Dobrava-Belgrade virus (DOBV)) and reservoir species (colonized *Microtus agrestis* and *Lemmus sibiricus*, wild-trapped *Myodes glareolus*, *Apodemus sylvaticus* and *Apodemus agrarius*), Klingstrom et al. [31] concluded that spill-over could occur and that the probability for a non-reservoir species to get infected with a given hantavirus increased gradually with its phylogenetic relationships with the natural reservoir species. As such, Forbes et al. [35] found that infection of lemmings with PUUV was highly unlikely.

Other experiments were dedicated to the analyzing of hantavirus infection in heterologous reservoir species. As such, McGuire et al. [54] have shown that Maporal virus (MAPV), a South American hantavirus whose reservoir is the delicate pigmy rice (*Oligoryzomys delicates*), was able to infect deer mice in laboratory conditions. Similarly, de Souza et al. [55] have revealed experimentally that Rio Mamore virus (RIOMV), another south American hantavirus naturally infecting *Oligoryzomys microtis*, was able to infect other *Sigmodontinae* rodents (*Necromys lasiurus* and *Akodon* sp.).

## 4. Immune Mechanisms behind the Biology of Reservoir/Hantavirus Interactions

### 4.1. The Effective Role of Antibodies during Infection Process

A first important issue that has been addressed experimentally to investigate the immune mechanisms behind reservoir/hantavirus interactions is the role of humoral antibodies in protecting against hantavirus infection.

#### 4.1.1. Maternal Antibodies

A large body of research has historically focused on the role of maternal hantavirus-specific antibodies. They have mainly been conducted on rats (e.g., [39,44,45,46]) and bank voles [13,25,82], using cross-fostering experiments in lab facilities or in outdoor enclosures. Less information is available for other reservoir/hantavirus species (however, see [27] and [62] for deer mouse/SNV). Altogether, these studies have shown that infected mothers transfer IgG and IgA to their offspring, transplacentally in utero and via breast feeding. These maternal antibodies protect neonatal rodents against hantavirus infection, during a transient period of about two and a half months for IgG (e.g., [82]) and less than three weeks for IgA [45]. However, as adults, these rodents become susceptible to hantaviruses and develop a persistent infection after being challenged with their specific virus [45,46]. Interestingly, the protection gained from maternal antibodies might not be complete, depending on the virus titer used during infection. Dohmae et al. [45] found that some neonatal rats could develop persistent infection despite maternal antibodies, when challenged with a high virus titer.

In adult reservoirs, the simultaneous observation of hantavirus and antibodies to hantavirus in infected animals has brought up questions about the protective role of non-maternal antibodies against hantavirus infection and about the mechanisms mediating persistence of hantaviruses in their natural reservoirs.

#### 4.1.2. Antibody Seroconversion

Experimental hantavirus infections have also enabled to characterize the nature and kinetics of antibody responses, including neutralizing antibodies that occur in the first two weeks of infection. Low titers of neutralizing antibodies can be detected at seven dpi [8,47]. During PUUV infection of wild bank voles, neutralizing antibodies were only found at 14 dpi [16]. Antibody response next increases and persist until the end of experiment [11,17]. Longer periods recorded include 150 dpi [14] or even 270 dpi [12]. Low levels of antibodies were even detected until 360 dpi in *Apodemus agrarius* infected with HTNV [11]. Results gathered from CMR surveys were consistent with those ones obtained in laboratory conditions, as PUUV infected bank voles monitored in the wild exhibited antibodies during the 15 months of the experiment [25].

### 4.2. Immune Mechanisms Mediating Hantavirus Persistence in Reservoir Hosts

Experimental studies have been critical to investigate the interaction of hantavirus with the host immune system and to emphasize the mechanisms enabling hantaviruses to establish persistent infections in their natural reservoirs. Indeed, natural population surveys usually do not enable to discriminate between immune patterns due to infection features (timing, strength, mode of transmission) or to phenotypic variability between individuals.

Two major hypotheses have been investigated from hantavirus/reservoir experiments, with regard to the mechanisms mediating hantavirus persistent infections. Hantaviruses could either suppress the natural reservoir immune responses that are required to resolve infection, or they could manipulate natural reservoir immune regulatory systems to overcome protective immune responses [41]. Immune response changes are surveyed during the course of hantavirus infection, usually from transcriptomic assays performed in organs of hantavirus replication (e.g., lungs) and sometimes also in immune organs, mainly the spleen. Globally, studies have shown that both hypotheses were met, but at different times of infection. Experiments have confirmed the role of innate immunity against hantavirus infection during the first time of the infection (acute infection), as previously established from in vitro cell line studies [49]. Briefly, the cell mediated immunity of reservoirs is enhanced following infection, as revealed by the increased expression of a number of pro-inflammatory and antiviral chemokines and cytokines in natural reservoirs (see for a review [83]). This is observed in the spleen of SEOV infected rats (IFN-g, IL4, CCl2, CCl5, e.g., [18,40]) and SNV infected deer mice (CCl2, CCl3, CCl5, IL12p35, IL21, IL23p19, [47]), but not (or in a lesser extent) in their lungs. This suggests that the lungs have less immunological activity during reservoir infections. Later in infection, the expression of genes encoding for pro-inflammatory cytokines is down-regulated in spleens and lungs [18], while the expression of regulatory factors increases in lungs (e.g., Foxp3, TGF-b, IL10 in rats, [41]). The immune response driven by regulatory T cells therefore seems to limit inflammation and resulting immunopathology, at the expense of virus persistence. This interpretation has been confirmed using functional inactivation of regulatory T cells of SEOV infected rats [41]. A similar pattern of regulatory T cell response predominance late in hantavirus infection seems to occur in deer mice infected with SNV, as shown by gene expression assessed from T cell culture derived from infected deer mouse spleen [49]. It would be interesting to investigate whether this balance between inflammation and immunopathology on one hand, and regulatory T cell responses on the other hand, is a general phenomenon driving reservoir/hantavirus interactions and resulting for evolutionary processes such as coadaptation.

Along this same line, some experiments have been designed to investigate whether hantavirus infections lead to similar or different responses in heterologous reservoir species compared to their natural reservoir. Cross-infections of deer mice have been conducted with ANDV and MAPV, with SNV infections being the reference model [53,54]. Some responses of deer mice were similar, regardless of the hantavirus used. In particular, the up-regulation of some chemokines and regulatory factors in the lungs and spleen is observed in both studies. But major differences were also observed. ANDV infection was transient as the virus was cleared within several weeks, due to the extremely high level of immune gene transcription observed in the spleen and to the strong CD4+ T-cell response mounted [53,83]. By contrast, MAPV infections seemed more similar to SNV infections, as the up-regulation of immune genes was, in both cases, low to modest in speen and lungs of deer mice. Unfortunately, it is still difficult to compare these studies and interpret the differences observed as the experimental protocols were very dissimilar (Table 1), and the response to MAPV infections was analyzed at 14 dpi only, which prevents from studying the persistence of the infection in the reservoir.

## 5. Challenges and Future Directions

### 5.1. Strong Individual Variation in Hantavirus Infection Histories: Challenges and Opportunities

The numerous reports of experimental hantavirus infection have highlighted the strong variability of infection histories, depending on the hantavirus/reservoir studied (see references in [51]), or on the geographic origin of reservoirs or hantaviruses [16,36]. This advocates for the need to enlarge hantavirus experiments to the wide array of biological models already described from field work. Only a large vision of the biology of hantavirus/reservoir interactions will enable to decipher the general mechanisms of persistent infection comprehensively [19]. More surprisingly, a strong variability of infection histories is also detected at the individual scale, even in the lab where standardized experimental conditions are applied.

#### 5.1.1. Sexual Dimorphism

Although ecological surveys have identified a large array of individual factors that could mediate variability of hantavirus/reservoir interactions, most of the experimental work done has focused on sex as a driver of inter-individual variability. Many studies have been carried on Norway rats and SEOV infections, and have shown a sexual dimorphism in the dynamics of SEOV replication [42]. Males shed the virus longer and via more routes [38,40]. They have more viral RNA copies present in target organs, such as the lungs, than females, during the persistent phase of infection [40,42]. Moreover, males are more aggressive and more likely to be infected with hantaviruses than females [20,38,40,41], which was also found in the long-tailed pygmy rice rat when exposed to ANDV virus [69]. However, a different pattern was observed in cotton rats, as Hutchinson et al. [51] found that males and females were equally susceptible to BCCV infection.

Experimental infections have been performed on Norway rats to decipher the mechanisms underlying this sexual dimorphism in reservoir response to hantavirus infections. Klein et al. [40] initially showed that IgG and Th1 responses differed between the sexes early after infection (males had higher IgG2a, IL-2, and IFNg concentrations in spleens than females) whereas Th2 responses (IgG1, IL4, IL10) did not. This higher level of Th1 response seems to be the result of increased virus replication in males [40]. Using molecular assays, Klein et al. [21] identified differentially expressed genes between males and females during the course of SNV infection, using lung tissue. They found a number of genes encoding for inflammatory and antiviral immune responses, antigen presentation and T cell responsiveness, cytokine receptors and transcriptional factors and antibody production that were up-regulated in females compared to males, during late stage of infection. These results indicate that females are more effective than males for mounting responses against SEOV in the lungs. Only genes encoding for heat shock proteins were up-regulated in males, indicating higher levels of cellular stress. Complementary experiments revealed that this sexual dimorphism was not mediated by sex steroid hormones only [84]. In particular, sex steroids were shown to act early in development (and not in adulthood) to affect the functioning of the innate and adaptive immune system later in life [38,84]. Whether this sexual dimorphism exists and is mediated by similar immune variations in other reservoir/hantavirus pairs remain to be investigated. Such knowledge could be an important input to improve our understanding of disease transmission and dynamics in natura (see for example [67]).

#### 5.1.2. Immune Phenotypes, Immunogenetic Profiles and Inter-Individual Variability in Sensibility to Hantavirus Infections

Beyond this sexual dimorphism, a number of hantavirus/reservoir experiments have revealed the presence of distinct phenotypes with regard to infection history. Such a pattern could not be identified from field surveys, as it could have resulted from differences in the timing of infection or in the probability of hantavirus encounter in the wild.

Laboratory experiments have highlighted some variability in the probability of a reservoir to be infected with his specific hantavirus, despite exposure with hantavirus (e.g., [32]). These studies, as well as the CMR surveys, have also shown some variability in the pattern of hantavirus replication (viral load and distribution) and excretion once a reservoir is infected (e.g., in PUUV/bank voles, [16,23,36,58], or in SNV/deer mice, [15,47]). More specifically, these studies reported two divergent phenotypes, one exhibiting restricted viral replication, with virus antigen being detected in a low number of organs (this was potentially associated with low levels of antigen expression, [47]). The second kind of phenotypes exhibited viral replication in many more organs and higher viral loads [15].

Immunogenetic polymorphisms within reservoir populations could mediate this variability in hantavirus/reservoir interaction outcomes. This hypothesis has not been addressed experimentally (however, see [85]), and natural population studies are still very scarce (however, see [86]). We advocate for more research on this issue, as it will provide important fundamental knowledge to better understand mechanisms mediating viral persistent infection, but also because this inter-individual heterogeneity has strong epidemiological consequences. These individuals exhibiting increased likelihood of being infected and higher excretion rate should be those with higher probability to transmit the virus and can be considered as super-spreaders [87]. Further research should help evaluating how host immune genotypes and phenotypes modulate reservoir response to hantavirus infection and, in turn, influence hantavirus epidemiology in the wild.

#### 5.1.3. Coinfections and Microbiome

It is now well established that pathogens within their hosts are not isolated but interact and evolve within a large community of parasites (*s.l.*) and microorganisms that can be commensal or pathogenic [88,89].

Interactions among co-infecting parasites may have important consequences for disease severity or disease epidemiology (e.g., [90]). Coinfection has therefore been highlighted as a potential reservoir feature that could mediate inter-individual heterogeneity with regard to infections. In particular, coinfections could be responsible for immune changes [90] that could lead to the simultaneous presence of different phenotypes within populations or infected animals, as described above.

A single experimental study has previously reported the impact of coinfection on hantavirus infections. Lehmer et al. [50] have shown that deer mice experimentally coinfected with SNV and the bacteria *Bartonella* exhibited different immune responses than mice infected with a single pathogen. This first study corroborates the hypothesis that coinfection may influence immune pathways and in turn, the probability of reservoirs to be infected with hantavirus, and to replicate and excrete the virus. The impact of coinfection on hantavirus/reservoir interaction is therefore an avenue of research that is worth exploring in the future. Several results gained from natural population surveys already provide the bases for making hypotheses to test experimentally, as for example the role of Th1/Th2 balance and the impact of gastrointestinal helminths on PUUV/bank vole interactions [91,92], or the influence of bacteria coinfecting reservoirs in natural populations [93].

An important extension of this multi-pathogen point of view concerns the new concept of pathobiome, which considers the pathogenic agent integrated within its biotic environment [89]. This concept incorporates the established influence of the commensal microbiome on host health and in particular on the regulation of animal immune responses. Thus, commensal microbiome diversity and composition might impact host susceptibility to disease and disease severity [94]. Therefore, considering microbial diversity within reservoirs as a potential driver of inter-individual variability in hantavirus infection outcomes should greatly improve our understanding of hantavirus infection histories and epidemiology. The recent advent of high throughput sequencing technologies now enables to describe microbial communities and offers a unique opportunity to address the issue of environment/microbiome/hantavirus/reservoir interactions, hence extending the conceptual framework proposed by Reusken and Heyman [95].

### 5.2. Viral Diversity and Evolution: under-Explored Factors in Understanding Hantavirus/Reservoir Interactions

#### 5.2.1. Adaptation during Virus Isolation

Hantaviruses are notoriously difficult to isolate. Many hantavirus isolates have been propagated on conventional cell lines, but there is only a very limited number of cell line adapted hantaviruses available for research. In particular, cell line adaptation of hantavirus is often performed on Vero E6 cells that lack the capacity to produce IFN-α/β. This has led to accumulation of adaptations to these cell lines and to the evolution of viral substrains with phenotypic properties that differ from those of the parental wild-type strain [96]. For example, Lundkvist et al. [97] have shown that PUUV cell culture has resulted in the accumulation of mutations in the S segment that limit the infectivity of PUUV Sotkamo strain in its natural reservoir animal. The PUUV strain Kazan-wt, which is not cell line adapted, seems to best mirror the situation in nature when used for experimental infection than a Kazan-E6 adapted strain [31,98]. Overall, most results gathered from PUUV experiments suggest that the adaptation of an isolate to cell culture growth can alter its infectivity for the natural reservoir. This probably reflects a change in the selective pressure when the virus was suddenly forced to reproduce in a new reservoir, namely, Vero E6 cells, which also allowed more variants to evolve. Similarly, Hutchinson et al. [51] have found that the passage history of SNV could affect whether the infections were transient or persistent. The potential adaptation of other hantaviruses (e.g., CDGV, SEOV) to Vero E6 cells and the consequences on the experimental results observed remains an open question in many studies [29,99].

#### 5.2.2. Viral Diversity and Quasispecies Dynamics

Because of the low fidelity of their RNA-dependent RNA polymerases, RNA viruses replicate as a set of close variants named viral quasispecies [100]. The resulting intra-host viral genetic diversity allows for a rapid evolution based on the selection of variants pre-existing in a mutant spectrum [97] and is an important factor of the virulence of RNA viruses. This diversity confers viruses with the ability to infect new hosts, evade the immune system (antigenic variation) and resist to drugs. Therefore, it constitutes a selective advantage to adapt to adverse growth conditions for the virus population [101]. However, despite the importance of these mechanisms for understanding the host-pathogens relationships, the study of intra-host genetic diversity strangely remains poorly explored for hantaviruses with the exception of a few studies carried in laboratory conditions.

The genetic diversity of hantavirus has been highlighted in early studies that have analyzed hantavirus isolate genomes. Chizhikov [102] has shown that a Sin Nombre Virus (SNV) isolate obtained from the lung homogenate of a deer mouse and passaged two times in laboratory-bred *P. maniculatus* and then five times in Vero E6 cells presented five (including one extra G in the noncoding region), eight and three nucleotide differences on the S, M and L genomic RNA segments respectively. Similarly, Lundkvist et al. [97] revealed that the viral populations from both wild-type (wt) (bank vole-passaged) and Vero E6 cell-cultured variants of PUUV Kazan strain included mixtures of closely related variants (quasispecies). They also reported an increased genetic heterogeneity of the viral populations during the first phase of the adaptation process (p0 to p3) on both studied genes (M and S). Interestingly, these studies have shown that the mutations occurring during hantavirus isolation did not result in alteration of deduced amino acid sequences of the N, GPC, and L proteins of SNV variants [102] neither in the consensus sequences from the wt and E6 PUUV Kazan strain. Indeed no difference was observed in the coding region of the S gene, whereas noncoding regions were found to be different at two positions [97].

Similar results have been found during experimental surveys of hantavirus infections. Feuer et al. [65] have analyzed SNV diversity and evolution during a mark-recapture study of deer mice. They have shown that 60% of all nucleotide substitutions occurring in the G1 region of SNV in naturally infected deer mice were non-synonymous. In laboratory conditions, Sironen et al. [33] have followed PUUV quasispecies dynamics during a PUUV infection experiment of naïve bank voles (recipients) using the nests of infected voles (donors). Two alleles of the PUUV S gene differing only by one silent mutation (A759G) were detected in the virus population used to infect the donor voles. The authors showed that this variant existed as a minority in the quasispecies swarms within the donor voles but repeatedly became fixed in the viral RNA quasispecies populations in the recipient animals during a single virus-transmission event. These studies emphasize the importance of silent mutations in hantaviruses that may be far from neutral and that could even contribute significantly to a general enhancement of replication and adaptation of these viruses to a changing environment [33]. Because the viral protein sequences seemed to be conserved during virus adaptation [97], some authors suggested that no genetic selection or adaptation occurred during virus growth in the experimentally infected laboratory rodents or in Vero E6 cells [102]. However, other authors suggested that silent mutations, conferring selective advantage that cannot arise on the level of protein structure, might involve codon-usage bias [103], as analyses showed that the resulting codon is more frequent in bank voles and humans, but rarer in PUUV [33].

Non-synomymous mutations also occurred, a single amino acid change in the glycoprotein of Hantaan virus (HTNV) virulent strain in a mouse model being sufficient to alter peripheral growth of the virus which affects invasion of the central nervous system of mice [104]. However, these results were not confirmed in another study investigating the effects of SNV strain diversity on transmission. Bagamian et al. [66] used naturally SNV-infected deer mice trapped in the field in a transmission experiment. Although these animals were from the same capture site, they yielded three different SNV S-segment variants (97.8% to 98.5% identity between sequences at the nucleotide level). However, despite this variability, no difference in transmission to susceptible deer mice within their respective enclosures was observed among these three viral variants. Moreover, all virus sequences retrieved from the experimentally infected deer mice were 100% identical to those of the suspected donor.

Another potential impact of hantavirus genetic diversity is related to spill-over. Viral diversity may enable hantaviruses to jump between different reservoir species and to emerge as new viral pathogens for new reservoirs. During their CMR study, Feuer et al. [65] showed that high levels of SNV genetic heterogeneity were detected in the salivary gland and bladder of infected deer mice. This finding could be of main importance with regard to spill-over, since hantavirus shedding and transmission between reservoirs or between reservoirs and human may occur through contaminated saliva or urine. SNV variants that are found in abundance in the bladder may increase the disease capacity of the virus: viral transmission to new reservoirs could be favored by the transfer of a variant, only present at low levels in an infected individual but presenting selective advantage for the spread and establishment into a new species. Up to now, this hypothesis has not been investigated yet, but it is likely that the combination of experimental infections and sequencing analyses would be critical to address this issue in the future.

#### 5.2.3. Viral Diversity, Persistence and Immune Escape

Experimental infections on natural reservoirs have also allowed investigating the role of hantavirus diversity in viral persistence and immune escape. This issue has mainly been addressed using the SNV/deer mouse pair. During their CMR studies, Feuer et al. [65] observed complex mixtures of SNV variants within one time point and over subsequent time points in both coding and noncoding regions of the genome. However, a single viral strain seemed to predominate in infected deer mice, probably as a result of a greater replicative capacity or fitness. Some nucleotides and amino acid positions were preferentially mutated, but no particular mutation became fixed in the SNV population over sequential time points before and after seroconversion. It was suggested that the emergence of new variants (fixation of a mutation) may occur after rare events, including the transmission of a minor variant to another deer mouse [65]. These results gathered from CMR surveys were quite different from what was observed in laboratory conditions [33], where fixation of mutation was observed during hantavirus transmission in recipient animals. Differences in experimental conditions and infection protocols could mediate further differences in hantavirus diversity and evolution.

Moreover, Feuer et al. [65] reported differences in mutation frequency between the organs examined. Higher SNV diversity was detected in the spleen, at both the nucleotide and amino acid levels, whereas low viral diversity was observed in the lungs and liver. It would be interesting to replicate this study to verify if this pattern can be extrapolated to other reservoir/hantavirus pairs, and to investigate whether this difference between spleen and lungs or liver is mediated by the major involvement of the spleen in the immune responses to blood-borne antigens. In this case, the enhancement of viral diversity in spleen could be favored as a mechanisms of viral immunological escape.

Exploring the role of viral diversity in the persistence of hantavirus in their reservoirs hence appears as an outstanding question that remains to be investigated in the future.

#### 5.2.4. Within Reservoir Hantavirus Diversity: New Insights from High Throughput Sequencing (HTS)

Exploring the genetic diversity of hantavirus population and its evolution within reservoirs is still a challenging task. However, it is interesting to note that high-throughput sequencing (HTS) is now commonly used for a large array of viruses studies (e.g., [105]) and is a critical opportunity for hantavirus research. These technologies are still scarcely used in the case of hantaviruses. Moreover, the rare studies that used HTS to analyze hantavirus diversity were often performed on patient, cell culture-based or wild reservoir samples [106,107,108,109]. To our knowledge, HTS has never been combined with experiments to assess hantavirus diversity and its evolution throughout the course of reservoir infection. It is likely that the ultra-low viral loads in experimentally reservoir samples (transient viremia, presence of viral RNA in the blood and organs below the limits of PCR detectability, [23]) prevents from developing HGS technologies on hantaviruses. This problem of detectability is particularly important with hantaviruses like PUUV as the level of viremia is considerably low [110,111]. Selective host RNA depletion and compensatory protocol adjustments for ultra-low RNA inputs could help bypassing these limitations in the future [112]. The combination of experimental viral evolution with HTS will then represent a powerful tool to identify and follow selected variants in controlled and replicable experimental settings and to reveal viral adaptive pathways [113,114,115].

## 6. Conclusions

In conclusion, we advocate for the need to continue to develop experimental approaches on hantaviruses and their natural reservoirs, at the interface of evolutionary ecology and virology, as in vitro studies and laboratory animal models experiments do not allow to fully investigate virus-reservoir interactions. In the future, it is important to fill the gaps with regard to the biological pairs that have been included in such studies. Many of hantavirus/reservoir pairs have been under-studied (especially soricomorphs and bats, but also rodents), which prevents the robust comparative analysis of the eco-epidemiological and evolutionary features assessed during these experiments. Moreover, we think that there are several emerging scientific issues that deserve to be addressed experimentally on hantaviruses and their natural reservoirs. These research areas should improve our understanding of the epidemiology of hantavirus associated diseases at a local geographical scale, and provide a more detailed assessment of hantavirus disease risk at larger scales.

## Figures and Tables

**Table viruses-11-00664-t001a:** (a) Laboratory conditions.

	Hantavirus		Reservoir			Experimental Design		Refs
Species	Viral Strain	Passage History	Species	Sex/Age	Mode of Infection	Virus Dose	Duration	
PUUV								
	Hällnäs	Passaged twice in laboratory-bred *C. glareolus*	*Myodes glareolus* (colony established in 1959)	Suckling (5 to 8 day-old) and weanling (4 to 8 week old)	intramuscular and intracerebral	10^3,5^ ID_50_	270 days	[12]
	Kazan 6 C.g		*My. glareolus*	-	intramuscular	100 ID_50_	13 months	[13]
	Kazan-wt	Passaged twice in colonized bank voles	*My. glareolus,* colonized *Microtus agrestis,*	-	subcutaneous	10^4^ bank vole ID_50_	21 days	[31]
			*Lemmus sibiricus*					
	Kazan-wt		*My. glareolus* (colony established in 1990)	-	subcutaneous	100 bank vole ID_50_	30 days	[32]
	Kazan-wt		Colonized *My. glareolus*	-	subcutaneous	200 bank vole ID_50_	133 days	[32]
	Kazan-wt		Colonized *My. glareolus*	-	subcutaneous	100 bank vole ID_50_	38 days	[33]
	Sotkamo	Cell culture adapted after numerous passages on Vero E6 strain	Laboratory *Mesocricetus auratus*	Males/Four week old and eight week old	subcutaneous	3300 ffu	70 days	[34]
	Sotkamo		Wild *L. lemmus*	-	1—Peat and bedding materials collected from PUUV-infected vole cages		1–4 weeks	[35]
					2—vole-soiled cages		2–8 weeks	
					3—intranasal (urine)		3–8 weeks	
	Sotkamo		Wild *My. glareolus*	-	subcutaneous	1.7 × 10^3^ ffu	35 days	[16]
	Sotkamo		Wild *My. glareolus*	-	subcutaneous	1.7 × 10^3^ ffu	55 days	[36]
	K27	Vero-E6 cells	Laboratory *Me. auratus*	Females/6–8 weeks	intragastrically	1	35 days	[37]
						pfu; 10,000 pfu or 10,000 pfu γ-irradiated		
						PUUV (3 × 10 6 rad)		
SEOV								
	SR-11	Four times in Vero E6 cells	Long Evans rats *(Rattus norvegicus)*	Males and females/70–80 days	-	10^-4^, 10^-3^, 10^−2,^ 10^2^, 10^4^ or 10^6^ pfu	40 days	[38]
	SR-11		SPF Wistar rats (*R. norvegicus*)	-	intraperitoneal		-	[39]
	SR-11		Long Evans rats (*R. norvegicus)*	Males and females/>70 days	intraperitoneal	10^4^ pfu	40 days	[21]
	SR-11		Long Evans rats (*R. norvegicus)*	Males and females/70–80 days	intraperitoneal	10^4^ pfu	40 days	[40]
	SR-11		Long Evans rats (*R*. *norvegicus*)	Males/60–70 days	intraperitoneal	10^4^ pfu	60 days	[18]
	SR-11		Long Evans rats (*R. norvegicus*)	Males/60–70 days	intraperitoneal	10^4^ pfu	40 days	[41]
	SR-11		Long Evans rats (*R. norvegicus*)	Males and females/60–70 days	intraperitoneal	10^4^ pfu	40 days	[42]
	Type B-1	Vero E6 cells	Laboratory rats (*R. norvegicus*)	Females/neonates	intraperitoneal	5 × 10^3^ or 5 × 10^5^ LD_50_	-	[43]
	Type B-1		Laboratory rats (*R. norvegicus*)	Female/neonates	intraperitoneal	5 × 10^3^ or 5 × 10^5^ LD_50_	-	[44]
	Type B-1		Laboratory rats (*R. norvegicus*)	Females/neonates	intraperitoneal	5 × 10^3^ or 5 × 10^5^ LD_50_	-	[45]
	Type B-1	-	Laboratory F344/Jcl rats (*R. norvegicus*)	2 to 3 months old	intraperitoneal	-	-	[46]
	KI-83-262	Seven days on Vero E6 cells	Wistar rats (*R. norvegicus*)	Newborn (<24 h) and 8 weeks old	intraperitoneal	6.6 × 10^3^ ffu	-	[24]
	KI-83-262		Wistar rats (*R. norvegicus*)	Newborn (<24 h) and 7 weeks old	intraperitoneal	6.6 × 10^3^ ffu	-	[17]
	80–39	Three times in Wistar rats and six times in Vero E6 cells	Inbred Lewis rats (*R. norvegicus*)	6, 10 or 21 days	intraperitoneal	10^6^ TCID_50_ of virus stock in 0.1 ml	84 days	[19]
HTNV								
	76–118	Eight passages in weanling rats and two in suckling rats	Wild *Apodemus agrarius*	Adults	intramuscular	10^5.9^ ID_50_/0,3 mL or 10^8,2^ ID_50_/0,3 mL	-	[11]
	76–118		Large array of species, including colonized and wild rodents	-	intramuscular or intralung	10^3,5^–10^5,2^ infectious units/mL	-	[11]
	76118		Fischer F 344/N rats (*R. norvegicus*)	Suckling (1 to 3 day old) and weanling (4 week old)	intramuscular and intracerebral	10^6^ and 10^8,5^ ID_50_	180 or 360 days	[22]
SNV								
	SN77734		*Peromyscus maniculatus* (outbred colony)	4 to 6 week-old	intramuscular	5 doses of ID_50_	217 days	[15]
	SN77734	In vivo within deer mice	*P. maniculatus* (outbred colony)	-	intramuscular	5 doses of ID_50_	120 days	[27]
	SN77734		Outbred *P. maniculatus*	Males and females/6 to 10 weeks	intramuscular	20 animal ID_50_	20 days	[47]
	SN77734		Colonized *P. maniculatus*	-	intramuscular	the equivalent of 2 × 10^5^ genome copies	6 weeks	[48]
	SN77734		*P. maniculatus*	Males and females/6 to 10 weeks	intramuscular	20 animal ID_50_	45 days	[49]
	SN77734		Wild *P. maniculatus*	-	-	-	-	[50]
BCCV								
	-	Three times in Vero E6 cells	Laboratory cotton rats (*Sigmodon hispidus*)	Males/four week-old	subcutaneous	1000 TCID_50_	150 days	[14]
	-	Three times in Vero E6 cells	Laboratory cotton rats (*S. hispidus*)	3 weeks to 4 months	subcutaneous	1000 TCID_50_	-	[51]
ANDV								
	ANDV-9717869	Vero E6 cells	Laboratory *Me. auratus*	Females/6 to 8 weeks old	intramuscular, intragastric, subcutaneous, intranasal		35 days	[52]
	ANDV-9717869		Outbred *P. maniculatus rufinus*	Males/6 to 12 weeks old	intramuscular	200 ffu	56 days	[53]
CDGV								
	VHV-574	Vero E6 cells	Laboratory *Sigmodon alstoni*	21 days	subcutaneous	3.1, 1.1, −0.9, or −2.9 log_10_ CCID_50_	54 days	[29]
TULV								
	Moravia/Ma5302V/9	-	*My. glareolus,* colonized *Mi. agrestis*	-	subcutaneous	2 × 10^3^ ffu	21 days	[31]
DOBV								
	Slovenia	-	*My. glareolus*		subcutaneous	5 × 10^3^ ffu	21 days	[31]
MAPV								
	HV 9702105	Vero E6 cells	*P. maniculatus*	Males and females/8–16 weeks	subcutaneous	104 TCID_50_	56 and 14 days	[54]
RIOMV								
	HTN-0007	Vero E6 cells	Wild *Sigmodontinae* rodents	-	intraperitoneal	10^3^ RNA copies	18 days	[55]

ID_50_: Median Infectious Dose; TCID_50_: Tissue culture Infective Dose; CCID_50_: median cell culture infectious dose; ffu: focus forming units; pfu: plaque-forming units.

**Table viruses-11-00664-t001b:** (b) Environmental conditions.

Hantavirus/Reservoir	Experimental Design	Duration	Refs
PUUV/*My. glareolus*			
	Capture-Mark-Recapture (CMR)	5 years	[25]
	CMR	7 years	[30]
	CMR	Three consecutive winters	[56]
	CMR	3 years	[57]
	CMR	16 months	[58]
SNV/*P. maniculatus*			
	CMR	2 years	[26]
	CMR	2 years	[28]
	CMR	15 years	[59]
	CMR	6 years	[60]
	CMR	Spring and fall	[61]
	CMR	7 months to 4 years	[62]
	CMR	5 years	[63]
	CMR	Spring and fall	[64]
	CMR	2 years	[65]
	Outdoor enclosure	1 year	[66]
	Outdoor enclosures	8 months	[67]
SNV/Rodent community			
	Outdoor enclosures	2 months	[68]
ANDV/Sigmodontine rodents			
	Outdoor enclosure	2 years	[69]
SEOV/*R. norvegicus*			
	CMR	2 years	[70]
	CMR	3 years	[71]
MRPV/*Zygodontomys brevicauda* and *Oligoryzomys fulvescens*	CMR	5 months	[72]

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
