# Peer review of "The Needs for Developing Experiments on Reservoirs in Hantavirus Research: Accomplishments, Challenges and Promises for the Future"

_viruses, 2019, doi:10.3390/v11070664_

Reviewer 1 Report

The paper is well written and contains a large number of experimental information on Hantaviruses collected from more than 100 scientific references. The information provided extensively covers multiple aspects of the pathogenesis of Hantaviruses infections focusing particularly on virus-host interaction.

Despite this, its reading is rather tiring and the conclusions are few, with the exclusion of the constant call for further research. To improve the work, I believe it can be reduced by at least a third, possibly citing only the most consolidated data and possibly summarizing them also in the conclusions, which I believe should be increased.

Minor point

Line 13 substitute “humans” with human infections

Line 31 substitute “people” with humans

Line 53 delete “recently”

Line 59 delete “these”

Line 67 delete, as redundant, “It is therefore important to synthesize the knowledge gathered from the experimental studies that have been conducted on hantaviruses and their natural reservoirs (both in the wild and in laboratory settings), considering the dynamics of infection, the mechanisms of hantavirus transmission and persistence in their reservoirs.”

Paragraph 2. Insights into infection dynamics of hantaviruses and their impact on reservoir fitness

Several repetitions of concepts, the text must be reduced

Page 7 line 8  explain “ dpi” here and not in line 17

             In discussion must be underlined that a weak point of this comparison on detection of viral RNA at different time could be due to the lack of standardization in molecular test with potential significant difference in sensibility.

Author Response

 The paper is well written and contains a large number of experimental information on Hantaviruses collected from more than 100 scientific references. The information provided extensively covers multiple aspects of the pathogenesis of Hantaviruses infections focusing particularly on virus-host interaction.

We thank the reviewer for this comment. We paid particular attention to cover the literature on hantavirus/natural reservoir  experiments that enabled to describe the infection dynamics of hantaviruses in their reservoirs, to analyse epidemiological features of these infections (transmission, pathogenesis …), and to learn more about the mechanisms mediating viral persistence in reservoirs and maintenance in natura.

Despite this, its reading is rather tiring and the conclusions are few, with the exclusion of the constant call for further research.

We agree that the concluding sentences calling for futher research may have been annoying. We improved the text in consequence.

To improve the work, I believe it can be reduced by at least a third, possibly citing only the most consolidated data and possibly summarizing them also in the conclusions, which I believe should be increased.

We reduced the main text to remove redondancies and information that were too trivial.

Nevertheless, we do not agree to cite only the most 'consolidated' data. One of the main interest and strength of this paper, by putting together all the experiments that have been published on this topics, is to emphasize the variability between individuals and between protocols in the patterns observed (epidemiology, evolution, mechanisms of persistence…). This variability may have important biological meaning and as such, it deserves attention. By focusing on ‘the most consolidated data’, we would turn a blind eye to this variability, and provide a wrong picture of hantavirus / reservoir interactions.

For these reasons, we decided to focus our conclusions on the gaps that need to be filled and on the opportunities that come from these comparisons between the experiments that have been published, rather than on a summary of conclusions derived from a few 'consolidated data'.

Minor point

Line 13 substitute “humans” with human infections

done

Line 31 substitute “people” with humans

done

Line 53 delete “recently”

done

Line 59 delete “these”

Line 67 delete, as redundant, “It is therefore important to synthesize the knowledge gathered from the experimental studies that have been conducted on hantaviruses and their natural reservoirs (both in the wild and in laboratory settings), considering the dynamics of infection, the mechanisms of hantavirus transmission and persistence in their reservoirs.”

done

Paragraph 2. Insights into infection dynamics of hantaviruses and their impact on reservoir fitness

Several repetitions of concepts, the text must be reduced

done

Page 7 line 8  explain “ dpi” here and not in line 17

done

             In discussion must be underlined that a weak point of this comparison on detection of viral RNA at different time could be due to the lack of standardization in molecular test with potential significant difference in sensibility. 

We agree that viral studies might not be comparable if they rely on different molecular tests with potential significant difference in sensibility. We have added a sentence in paragraph 2 on this point.

Reviewer 2 Report

Dr. Carbonnel and colleagues describe the current state of research for interface between animals and hantaviruses, including animal models.  Overall this is a good manuscript and worthy of publication.  It would be improved if the authors spent some additional time discussing the interpretation of research, rather than just the data.

Minor comments:

Line 302: Limit on animal numbers at ABSL-4 is not correct.  Rather, it would be better to stress how few laboratories can perform the work.

Adaptation discussion is placed in the difficulties section.  This would be appropriate for its own section.

Author Response

Dr. Charbonnel and colleagues describe the current state of research for interface between animals and hantaviruses, including animal models.  Overall this is a good manuscript and worthy of publication.  It would be improved if the authors spent some additional time discussing the interpretation of research, rather than just the data.

We thank the reviewer for this positive comment. We understand that it can be frustrating that we provide information on the experiments performed and data gathered and future needs or opportunities, but not on interpretation of data in terms of epidemiology or evolution… However, we should stress that i) this review is already quite long and dense with information, ii) the reviewer 1 asked for a strong reduction in length, and iii) there are already nice reviews published that focused on interpretation of research (e.g. Forbes et al. 2018 ;Schountz et al. 2014…). We therefore decided to keep the manuscript focused on experiments, data, gaps and opportunities.

Minor comments:

Line 302: Limit on animal numbers at ABSL-4 is not correct.  Rather, it would be better to stress how few laboratories can perform the work.

We agree with this comment. Because the text had to be reduced on the recommendations of reviewer 1, this part was removed from the manuscript (too commonplace information).

Adaptation discussion is placed in the difficulties section.  This would be appropriate for its own section.

We include this part in its own sub-section (5.2.1).